# Isolation, Diversity and Characterization of Ulvan-Degrading Bacteria Isolated from Marine Environments

**DOI:** 10.3390/molecules27113420

**Published:** 2022-05-25

**Authors:** Reiji Tanaka, Yu Kurishiba, Hideo Miyake, Toshiyuki Shibata

**Affiliations:** 1Graduate School of Bioresource, Mie University, 1577 Kurimamachiya-cho, Tsu 514-8507, Mie, Japan; 518525@m.mie-u.ac.jp (Y.K.); miyake@bio.mie-u.ac.jp (H.M.); shibata@bio.mie-u.ac.jp (T.S.); 2Seaweed Biorefinery Research Center, Mie University, 1577 Kurimamachiya-cho, Tsu 514-8507, Mie, Japan

**Keywords:** ulvan, ulva, degradation, lyase, bacteria, *paraglaciecola*, *vibrio*, *echinicola*, *algibacter*

## Abstract

In this study, we aimed to isolate bacteria capable of degrading the polysaccharide ulvan from the green algae *Ulva* sp. (Chlorophyta, Ulvales, Ulvaceae) in marine environments. We isolated 13 ulvan-degrading bacteria and observed high diversity at the genus level. Further, the genera *Paraglaciecola*, *Vibrio*, *Echinicola*, and *Algibacter*, which can degrade ulvan, were successfully isolated for the first time from marine environments. Among the 13 isolates, only one isolate (*Echinicola* sp.) showed the ability not only to produce externally expressed ulvan lyase, but also to be periplasmic or on the cell surface. From the results of the full-genome analysis, lyase was presumed to be a member of the PL25 (BNR4) family of ulvan lyases, and the bacterium also contained the sequence for glycoside hydrolase (GH43, GH78 and GH88), which is characteristic of other ulvan-degrading bacteria. Notably, this bacterium has a unique ulvan lyase gene not previously reported.

## 1. Introduction

Seaweed has been attracting attention as a novel biomass resource, and has been proposed as a raw material for sustainable biomass energy and biomaterials to replace terrestrial biomass in the future [1,2]. The advantages of seaweed as a biomass resource are that it does not require agricultural land or water for aquaculture, and the yield per unit area is higher than that of land crops such as corn [3]. The market for seaweed as a food is limited to Japan and some other areas, and thus, competition with food resources is suggested to show less impact than for land crops. Additionally, as Japan is a marine resource nation (with the 6th largest sea area in the world) and has a long history of seaweed farming with established technology, there is potential to develop large-scale marine aquaculture in the future. However, large algae, such as kelp and red algae, are members of a community known as the underwater forest. This forest is ecologically valuable because it provides food and habitats to marine life, and allowing harvesting can negatively impact the populations of marine life that inhabit these areas [4].

In recent years, large amounts of green algae (*Ulva* sp.) have appeared along the coast of Japan, e.g., Tokyo Bay or Hakata Bay [5]. When green algae form green tides along the coast, resulting from an over-proliferation of the algae, it causes various problems such as disruption of the ecosystem, the production of a foul odor following its decay, and damage to the landscape. The algae are collected and sent to landfill, or utilized as biomass or fertilizer, etc.; however, there are significant associated costs, necessitating the study of additional effective utilization methods. On the other hand, green algae contain functional polysaccharides with potential applications. Specifically, ulvan is a water-soluble intercellular mucilage polysaccharide contained in green algae such as ulva, and accounts for 8–29% of the dry weight of green algae [6]. Ulvan is comprised of repeating disaccharides, in which glucuronic acid, iduronic acid, and xylose is bound to 3-sulfated rhamnose [7,8,9]. The 3-sulfated rhamnose is thought to have antiviral, antitumor, and blood-cholesterol-lowering effects [10], while iduronic acid, a rare sugar, can be applied to the synthesis of heparinoids [11]. In addition, since ulvan constitutes various disaccharides, it can be used in functional foods and pharmaceuticals following depolymerization [12,13,14,15,16,17,18]. Acid hydrolysis and enzymatic degradation are mainly employed to generate low-molecular-weight polysaccharides. However, acid hydrolysis exhibits strong decomposing power and has some drawbacks, such as destruction of the sugar itself and inconsistent molecular weights of oligosaccharides, disaccharides, and monosaccharides. Therefore, gradual or mild decomposition of polysaccharides using ulvan lyase, which is an enzyme contained in marine bacteria, is required [19]. However, few studies have focused on ulvan-degrading bacteria and the isolation of ulvan lyase. Table 1 lists previously isolated ulvan-degrading bacteria, and ulvan-degrading activity has been reported for *Nonlabens ulvanivorans* [20], *Pseudoalteromonas* sp., *Alteromonas* sp. [21], *Formosa agariphila* [22], and *Glaciecola* sp. [23]. *N. ulvanivorans*, *Pseudoalteromonas* sp. and *Alteromonas* sp. were isolated from the gastrointestinal tract of *Aplysia* sp., nudibranchs, and some shrimps, respectively, suggesting that these bacteria are associated with green algae feeders.

In this study, we aimed to comprehensively isolate bacteria with ulvan-degrading activities, using decomposed ulva and concentrated seawater as isolation sources. The diversity of the isolated ulvan-degrading bacteria was confirmed using phylogenetic analysis. Subsequently, the bacteria were screened for ulvan-degrading activity, and 13 isolates were selected. The degradation activity of isolates was measured using a UV absorbance method. Among the 13 isolated ulvan-degrading bacteria, *Echinicola* sp. 20G, which showed a different degrading activity from other bacteria, was subjected to full-genome analysis, and polysaccharide utilization loci related to ulvan degradation were identified.

## 2. Materials and Methods

### 2.1. Preparation of Ulvan

In April 2018, *Ulva* sp. (Ulvaceae, Ulvales, Chlorophyta) were collected at Owase Bay, Mie Prefecture, Japan. After drying with an electric fan, the algae were cut into appropriately sized pieces, placed in a box with a desiccant (silica gel), and dried for one week. Next, the sample was powdered and stored in the freezer at −30 °C as the powdered ulva sample. Ulvan was extracted from the powdered sample using ethanol precipitation. In brief, 10 g of powdered ulva and 120 mL of distilled water were mixed for 1 h, then the solution was centrifuged at 3300× *g* for 5 min. The supernatant was transferred to a Falcon tube, and 1% of the supernatant was added with 3 M sodium acetate and 2 times the amount of ethanol. The tube was slowly inverted to precipitate the white ulvan, then the precipitated ulvan was pelleted using centrifugation (3300× *g* for 10 min) and the supernatant was discarded. The pellet was collected and dried, then it was re-dissolved in distilled water, frozen, and lyophilized.

### 2.2. Sample Collection and Preparation

The sources of isolation were decomposed ulva collected on the Machiya coast (May 2018, sample name: uL), surface seawater from the Machiya coast (September 2019, sample name: M), Mikawa Bay point MK1 bottom sediment (May 2019 sample name: D), Mikawa Bay point MK1 bottom seawater (May 2019 sample name: 1) and Mikawa Bay point MK6 bottom seawater (May 2019 sample name: 6). The samples from MK point were collected by SEISUIMARU, a training ship of Mie University. Sample details are listed in Table 2.

For the preparation of ulva samples (sample name: ul), ulva were placed in a 2.0 mL tube together with a 3.0 mm iron ball and sterilized artificial seawater, and crushed for 30 s using a bead crusher (TAITEC, Japan). For the concentrated seawater samples, 1 L of seawater was filtered using a GFC filter (pore size 1.2 μm) and a 0.2 μm nucleopore membrane filter. After filtering with the GFC filter (sample name: G), filtration was performed with the 0.2 μm nucleopore membrane filter (sample name: N).

### 2.3. Isolation of Ulvan-Degrading Bacteria from Marine Environments

Purified decomposed ulva samples (sample name: ul) were inoculated into 1/10 Marine broth (BD-Difco, Sparks, MD, USA) with 0.5% ulvan and cultured at 25 °C for 5, 9, and 20 days. The other samples (sample names: M, D, 1, and 6) were cultured at 25 °C for 10 to 20 days. 

Each culture solution was streaked onto Marine agar plates (BD-Difco, Sparks, MD, USA) and cultured at 25 °C for 2–3 days, then random colonies were picked and cultured on Marine agar plates at 25 °C for 2–3 days. Several colonies that were expected to be different from each other were selected based on their color, shape, and size. When new colonies were confirmed at a later date, additional pure isolation was performed. The colonies were subcultured on Marine agar plates at 25 °C for 2–3 days, then subcultured on Marine agar slope media and stored as isolates at 25 °C.

### 2.4. Establishment of a Screening Method for Ulvan-Degrading Activity

Cetylpyridinium chloride (CPC) was used for screening of ulvan-degrading activity. CPC is known to react with acidic polysaccharides and turn cloudy (23). A 2-mL aliquot of 5% CPC solution was added dropwise to the Marine agar plates with and without 0.5% ulvan. After immersion with CPC, plates were incubated at 25 °C for 30 min and changes in each medium were confirmed (Figure 1).

### 2.5. Ulvan-Degrading Activity

Ulvan-degrading activity was assessed according to the size of the halos using the CPC method [23]. Thirteen isolates with ulvan-degrading activity were infused and cultured on 1/10 Marine agar plates supplemented with 0.5% ulvan (25 °C, 120 h). Decomposition was confirmed as halos followed using above mentioned method (Section 2.4), and the diameter of the halos were measured.

### 2.6. Phylogenetic Analysis Based on 16S rRNA Gene Sequences

DNA was extracted from individual isolates with ulvan-degrading activity using a DNA extraction kit (Promega, Madison, WI, USA). Amounts of 1-μL of 4 μM primers 8F (5′-AGAGTTTGATCCTGGCTCAG-3′) and 1492R (5′-GGCTACCTTGTTACGACTT-3′), and 0.25 μL of TaKaRa Ex Taq polymerase (TaKaRa, Tokyo, Japan) were used. A 1-μL aliquot of DNA was added to each sample, (totally, 20 μL) and PCR was performed with a thermal cycler (i-Cycler, Bio-Rad, Hercules, CA, USA) under the following conditions. After one cycle of heat denaturation at 95 °C for 4 min, further heat denaturation at 95 °C for 30 s, annealing at 55 °C for 30 s, and an extension reaction at 72 °C for 1.5 min were performed for 25 cycles. The obtained PCR product was subjected to 1.5% agarose gel electrophoresis, and the appearance of a single band of about 1.5 kbp was confirmed. Sequencing was performed using a BigDye Terminator v3.1 Cycle Sequencing Kit (Applied Biosystems, Wolsam, MA, USA).

The nucleotide sequence obtained by sequencing was edited using the gene analysis software Chromas to determine the 16S rRNA gene sequence. Based on the sequence, a nucleotide sequence homology search was performed using the Basic Local Alignment Search Tool (BLAST) of the National Center for Biotechnology Information (NCBI) (accessed on: 1 February 2022), and the nucleotide sequences of closely related species were obtained. Next, we prepared a phylogenetic tree using the multiple alignment software MEGA version 7.0 (https://www.megasoftware.net/ accessed on 1 February 2022). Multiple alignment was performed, and 1000 bootstrap tests were performed using the Neighbor Joining (NJ) method. 16S rRNA gene sequences are listed in the Data Availability Statement section.

### 2.7. Determination of Degradation Activity by Absorbance Measurement

*Alteromonas* sp. ul20-2, *Echinicola* sp. 20G, *Vibrio* sp. 10 N and *Vibrio halioticoli* strain IAM14596 (this bacterium does not have degrading activity of ulvan and was used as negative control) were pre-cultured in Marine broth (25 °C, 24 h). A 100-μL aliquot of the pre-culture was inoculated into 10 mL of 1/10 Marine broth containing 0.5% ulvan, and swirl culture was performed (25 °C, 72 h). The entire culture solution was transferred to a 15 mL Falcon tube and centrifuged (5800× *g*, 10 min) to separate the cells. The supernatant was concentrated 10-fold with a 3000 MW cut-off filter (Amicon Ultra-15 Centrifugal Filter Devices, Merck, Tullagreen, Ireland), and this was used as the extracellular concentrate. In addition, 600 µL of PBS was added to the bacterial cells as the precipitate, and ultrasonic crushing (THU-80, AS ONE, Tokyo, Japan) was performed on ice (40 W, 30 s, 3 times). The supernatant was obtained through centrifugation (5800× *g*, 10 min) and used as the cell-crushed solution.

Each sample was measured by adding 5 μL of the extracted fractions (extracellular concentrate or cell-crushed solution) to 95 μL of 0.05% ulvan solution. Absorbance measurements were performed (Infinite 200 PRO, TECAN, Mannedorf, Switzerland) at an absorbance of 235 nm every 1 min for 30 min at 30 °C, to observe changes over time. Values of absorbance of 235 nm indicate an absorption peak in the double bond of unsaturated uronic acid that is produced when ulvan is decomposed by ulvan lyase [22].

### 2.8. Identification of Ulvan-Degrading Lyase Gene Using Full-Genome Analysis of Echinicola sp. 20G

*Echinicola* sp. 20G was subjected to full-genome analysis. A sequence library was generated using the DNA extracted with the Promega DNA extraction kit. The long read was acquired using GridION X5 (Oxford Nanopore Technologies, Oxford, UK) and the short read was acquired using DNBSEQ-400 (MGI, Shenzhen, China). The obtained sequencing analysis results were hybrid-assembled under the default conditions of Unicycler. The assembly results were annotated with RAST (Rapid Annotations using Subsystems Technology), and the annotated genome was confirmed with SEED Viewer (https://pubseed.theseed.org/; accessed on: 1 February 2022). The genome sequences and raw sequences are listed in the Data Availability Statement section.

## 3. Results

### 3.1. Isolation of Ulvan-Degrading Bacteria

Bacterial isolation was performed once from concentrated seawater during the culture periods of 10 and 20 days. After culturing for a few days, new colonies that appeared to be different were selected and isolated. A total of 1 out of 44 isolates from the decomposed ulva (strain ul20-2) and 12 out of 135 isolates from the concentrated seawater were selected as ulvan-degrading bacteria (Table 2). Thus, 13 ulvan-degrading bacteria out of a total of 179 isolates were obtained.

### 3.2. Phylogenetic Analysis of Isolates Using 16S rRNA Gene Sequences

Figure 2 shows a phylogenetic tree prepared using 16S rRNA gene sequences of the 13 isolated ulvan-degrading bacteria. Of the 13 ulvan-degrading bacteria isolated (shown in red letters), seven of three genera of *Alteromonas* sp. N2, ul20-2, N3; *Pseudoalteromonas* sp. DN-1, DN-3, 6G-2; and *Nonlabens* sp. DG-3 are known ulvan-degrading bacteria. *Paraglaciecola* sp. MN-1, MN-2; *Vibrio* sp. 10N; *Echinicola* sp. 20G; and *Algibacter* sp. are novel ulvan-degrading bacteria. The isolated ulvan-degrading bacteria belong to the CFB phylum group and Proteobacteria (Figure 2).

### 3.3. Ulvan Degradation Activity

The ulvan-degrading activity of the 13 isolates were measured and the results are shown in Figure 3. The *P. atlantica* AR06 strain (NBRC 101679) stocked in our laboratory was used as a positive ulvan-degrading strain. The size of the decomposition spots (halos) of most isolates was similar, 20–30 mm, whereas the size of halos for *Echinicola* sp. 20G was 9 mm, which was slightly smaller than the others. Figure 4 shows the results of reaction time course of ulvan and the extracellular concentrate solution. The absorbance of *Alteromonas* sp. ul20-2 and *Vibrio* sp. 10N increased with time, while that of *Echinicola* sp. 20G increased very slightly, and the negative control remained almost unchanged. On the other hand, for the cell-crushed solution shown in Figure 5, the absorbance of *Alteromonas* sp. ul20-2, *Vibrio* sp. 10N and the negative control remained almost unchanged over time, while that of *Echinicola* sp. 20G highly increased in a time-dependent manner.

### 3.4. Identification of Genes Related to Ulvan Degradation in Echinicola sp. 20G

From the full genome information of *Echinicola* sp. 20G (annotated using RAST), polysaccharide utilization loci (PUL) related to ulvan lyase were observed in two different regions. PUL1 (shown in Figure 6) contained the glycoside hydrolase (GH) family, GH43 (CDS nos. 431, 442, 443). Several more ulvan-degrading rerated genes such as maltodextrin glucosidase (EC 3.2.1.20) (CDS no. 432), glucuronyl hydrolase (CDS no. 433), laminarinase (EC 3.2.1.39) (CDS no. 436), two xylan 1, 4-β-xylosidase (EC 3.2.1.37) (CDS nos. 441, 444), β-galactosidase (EC 3.2.1.23) (CDS no. 445), and TonB-dependent transporter (CDS nos. 439, 446) were confirmed.

PUL2 (shown in Figure 7) contained a rhamnogalacturonan lyase (CDS no. 1735). It was confirmed that the GH family glycoside hydrolases GH88 (CDS no. 1736) and GH2 (CDS no. 1760) were contained around this lyase. In addition, several genes involved in ulvan degradation, rhamnogalacturonide degradation protein RhiN (CDS no. 1716), two α-L-rhamnosidases (GH78) (EC.3.2.1.40) (CDS nos. 1743, 1751), five β-galactosidases (EC.3.2.1.23) (CDS nos. 1741, 1746, 1752, 1758, 1759), α-L-arabinofuranosidase (EC.3.2.1.55) (CDS no. 1740), two oxidoreductases (CDS nos. 1749, 1753), reductase (CDS no. 1789), and TonB-dependent transporter (CDS nos. 1728, 1732) were confirmed. Both clusters contained a large amount of glycoside hydrolase, but sulfatase could not be confirmed. Of the two ulvan-degradation-related gene groups confirmed in this study, CDS no. 434 of PUL 1 showed high homology with a PL25 (BNR4-like) protein, which is known as an endo-type ulvan lyase. The CDS no. 1735 of PUL 2 contained a lyase-like sequence, suggesting that it is highly likely to be a novel ulvan lyase.

## 4. Discussion

This study is the first to report the isolation of ulvan-degrading bacteria from decomposed ulva and concentrated coastal seawater for the purpose of comprehensively investigating the distribution of ulvan-degrading bacteria. Previous studies reported that ulvan-degrading bacteria were mainly isolated from organisms that feed on green algae. However, we comprehensively investigated ulva for the presence of bacteria that use ulvan as the sole carbon source. The filtration method used for obtaining seawater at different sites or water levels and seasons also raised the possibility of isolating ulvan-degrading bacteria. In addition, we successfully isolated bacteria from GFC-filtrated seawater samples, and it is suggested that ulvan-degrading bacteria were attached to particles that could not pass through the GFC filter (1.2 μm). This represents a new finding, in which ulvan-degrading bacteria were isolated from the green algae as well as such adherent fractions. In the phylogenetic analysis of 16S rRNA gene sequences of the 13 isolates, we successfully characterized *Paraglaciecola* [24], *Vibrio* [25], *Echinicola* [26], and *Algibacter* [27] as ulvan-degrading bacteria. These bacteria are found in the Proteobacteria and CFB groups, suggesting that ulvan-degrading bacteria widely inhabit seawater and are distributed in various bacterial groups.

In assessing degradation activity, the reaction kinetics of the extracellular concentrate of the *Alteromonas* sp. ul20-2 strain or *Vibrio* sp. 10N strain and ulvan showed an increase in absorbance of 235 nm, and these two isolates are suggested to have extracellular-type ulvan-degrading enzymes secreted to the outside of cells. On the other hand, the reaction kinetics of the cell-crushed solution of the *Echinicola* sp. 20G strain and ulvan showed an increase in absorbance of 235 nm. In addition, since the result of the ulvan degradation measurement for the *Echinicola* sp. 20G strain was weaker (a halo of only 9 mm) than those of the other bacteria (halo > 20 mm), the degradation mode is thought to differ from the existing extracellular degradation mode. In light of the potential applicability of this bacterium, we subsequently obtained the full genome of the *Echinicola* sp. 20G strain and performed a homology analysis with known ulvan lyases. As a result, we identified two ulvan lyases. One belongs to the well-known BNR4 family, while the other is presumed to be a previously unknown ulvan-degrading enzyme, and it is proposed to have a novel degradation mechanism from the ulvan-degrading bacteria identified to date. We discovered two ulvan lyases and four glycoside hydrolases (GH2, GH43, GH78 and GH88). We predict that these enzymes are enough to produce mono-ulvan saccharide based on previous work [4], and predictions of the pathway in *Echinicola* sp. 20G are shown in Figure 8. 

At present, 13 genes of known ulvan lyases are registered in the Carbohydrate-Active enZYmes (CAZy) database (www.cazy.org), and are classified into five polysaccharide lyases (PL24, PL25, PL28, PL37, PL40) [8,20,28,29,30,31,32]. Ulvan lyases of known mechanism of action (shown in Table 1) are all endo-type lyases secreted outside the cells and can only decompose hexasaccharides to disaccharides [8,9,20,23,28,30,31,32,33,34,35,36,37]. We predicted that the *Echinicola* sp. strain 20G has any possible N-terminal signal peptides for the enzymes using the SignalP program, and found signal peptides in both CDS no. 434 and CDS no. 1735 (data not shown). Nevertheless, this bacterium exhibits ulvan-degrading activity in cell-crushed solutions, suggesting that this bacterium may not only secrete an extracellularly expressed enzyme that degrades ulvan, but may also be periplasmic or cell-surface-associated, bound to a cellulosome-like structure. In the future, it is necessary to conduct protein expression analyses and clarify the mechanism responsible for ulvan degradation within cells in this bacterium.

## 5. Conclusions

We isolated 13 isolates of ulvan-degrading bacteria, and successfully isolated *Paraglaciecola*, *Vibrio*, *Echinicola*, and *Algibacter* for the first time. Only one (*Echinicola* sp.) of the 13 isolates showed the ability to produce ulvan lyase. From full-genome analysis, we assumed that lyase is a member of the PL25 (BNR4) family of ulvan lyases; moreover, we assumed that *Echinicola* sp. has a unique ulvan lyase gene that has not been previously reported and also contains the sequence for glycoside hydrolase, which is characteristic of other ulvan-degrading bacteria.

## Figures and Tables

**Figure 1 molecules-27-03420-f001:**
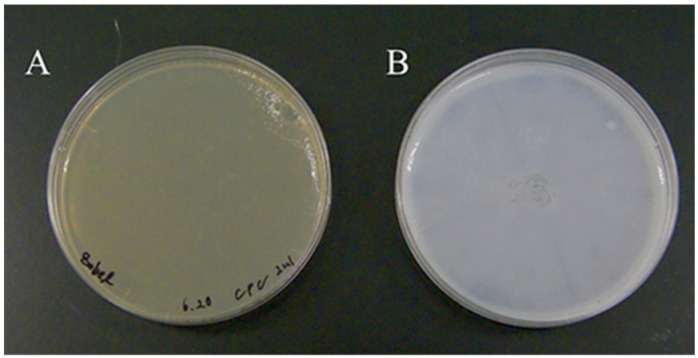
CPC reaction with ulvan. To identify ulvan degradation, 2 mL of 5% CPC solution was added dropwise to the Marine agar plates with and without 0.5% ulvan (**A**, Marine agar medium; **B**, Marine agar medium with 0.5% ulvan). (**A**) No change in color was observed; (**B**) the appearance of a white color indicated presence of ulvan. Ulvan-degrading activity cannot be indicated on such a test plate without bacteria. Instead, ulvan-degrading activity was indicated by formation of a clear zone around a colony that was able to degrade ulvan.

**Figure 2 molecules-27-03420-f002:**
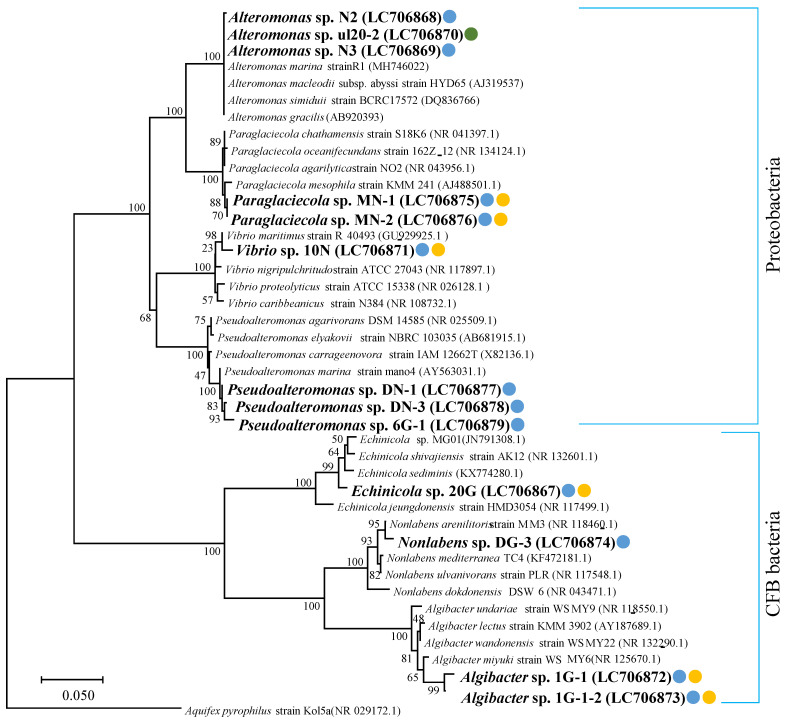
Maximum parsimony tree based on 16S rRNA gene sequences showing the relationship between ulvan-degrading bacteria and other bacteria. Bootstrap values (%) based on 1000 replications are shown at tree nodes. The sequence from this study is shown in bold text. Isolates from ulva are indicated by green circle. Isolates from seawater are indicated by blue circles. Novel ulvan-degrading bacteria are indicated by yellow circles.

**Figure 3 molecules-27-03420-f003:**
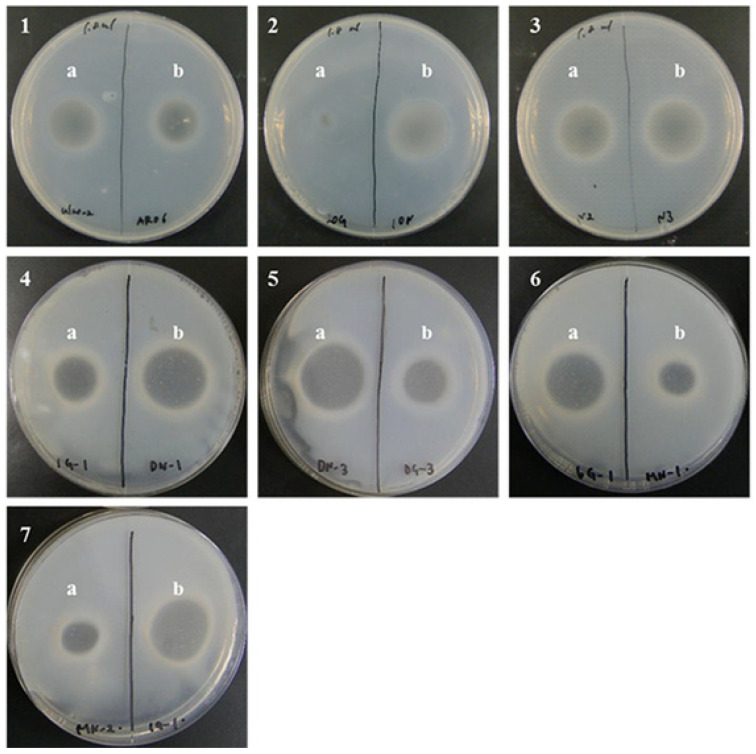
Determination of ulvan-degrading ability in bacteria. The 13 isolates of ulvan-degrading bacteria were cultured in medium containing ulvan for five days, and then immersed in CPC solution. **1a**, *Alteromonas* sp. ul20-2; **1b**, *Pseudoalteromonas* sp. AR06; **2a**, *Echinicola* sp. 20G; **2b**, *Vibrio* sp. 10N; **3a**, *Alteromonas* sp. N2; **3b**, *Alteromonas* sp. N3; **4a**, *Algibacter* sp. 1G-1; **4b**, *Pseudoalteromonas* sp. DN-1; **5a**, *Pseudoalteromonas* sp. DN-3; **5b**, *Nonlabens* sp. DG-3; **6a**, *Pseudoalteromonas* sp. 6G-1; **6b**, *Paraglaciecola* sp. MN-1; **7a**, *Paraglaciecola* sp. MN-2; **7b**, *Algibacter* sp. 1G-1-2.

**Figure 4 molecules-27-03420-f004:**
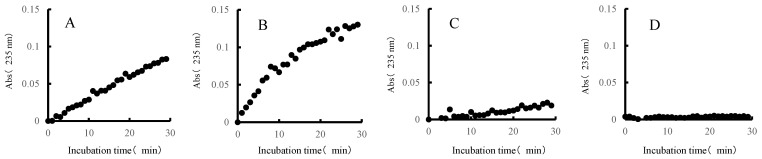
Reaction time course (235 nm) of ulvan and extracellular concentrates ((**A**), *Alteromonas* sp. ul20-2; (**B**), *Vibrio* sp. 10N; (**C**), *Echinicola* sp. 20G; (**D**), negative control).

**Figure 5 molecules-27-03420-f005:**
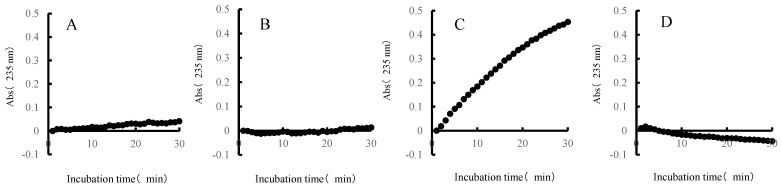
Reaction time course (235 nm) of ulvan and cell-clashed solutions ((**A**), *Alteromonas* sp. ul20-2; (**B**), *Vibrio* sp. 10N; (**C**), *Echinicola* sp. 20G; (**D**), negative control).

**Figure 6 molecules-27-03420-f006:**
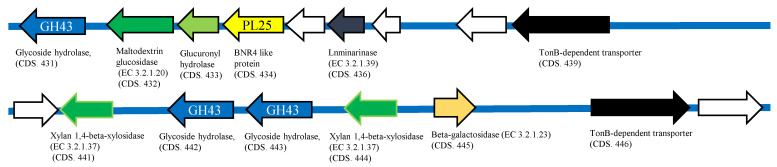
PUL1 of a putative ulvan-degrading-related gene of *Echinicola* sp. 20G. White arrows indicate hypothetical protein and gray arrows indicate genes that are not relevant to ulvan degradation.

**Figure 7 molecules-27-03420-f007:**
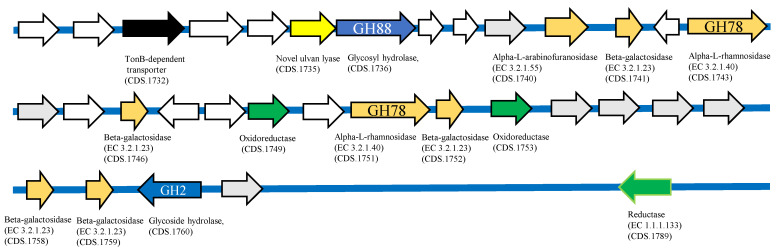
PUL 2 of a putative ulvan-degrading-related gene of *Echinicola* sp. 20G. White arrows indicate hypothetical protein and gray arrows indicate genes that are not relevant to ulvan degradation.

**Figure 8 molecules-27-03420-f008:**
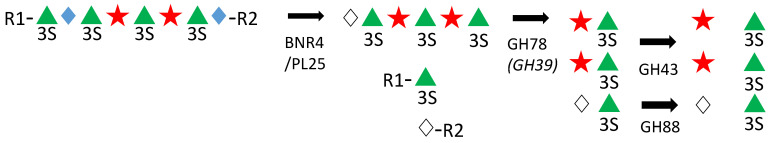
Model of the ulvan degradation pathway in *Echinicola* sp. 20G as suggested by the genomic and biochemical analyses in this study. This figure is modified based on Reisky et al., 2019 [4]. ▲, *L*-Rhamnose; ♦, *D*-Glucuronic acid or *L*-Iduronic acid; ♢, unsaturated uronic acid; ★, *D*-Xylose. Italic, not found in this study.

**Table 1 molecules-27-03420-t001:** List of previously isolated ulvan-degrading bacteria and polysaccharide lyase enzyme family or degradation products from ulvan.

Source	PL Family	Products	Product Composition
*Alteromonas* sp. KUL 17	PL24	DP2,4,6	-
*Glaciecola* sp. KUL 10	PL24	-	-
*Alteromonas* sp. KUL 42	PL24	-	-
*Alteromonas* sp. LOR 107	PL24	DP2,4	∆UA-R3S
*Pseudoalteromonas* sp. PLSV 3875	PL24	DP2,4	∆UA-R3S-IdoA-R3S
*Pseudoalteromonas*. PLSV 3925	PL24	DP2,4	∆UA-R3S-Xyl-R3S
*Alteromonas* sp. LOR 61	PL24	DP2,4	
*Pseudoalteromonas* sp. PLSV 3936	PL25	DP2,4	∆UA-R3S
*Alteromonas* sp. LOR 29	PL25	DP2,4	∆UA-R3S-Xyl-R3S
*Nonlabens ulvanivorans* NLR 492	PL25	DP2,4	
*Alteromonas* sp. A 321	PL25	DP2,4	∆GlcA-Rha3S, ∆GlcA-Rha3S-Xyl-Rha3S, ∆-Rha3S
*Formosa agariphila* KMM 3901	PL28	DP2-6	R3S-xyl-R3S, ∆-Rha3S-Xyl-Rha3S, ∆-Rha3S-xyl-Rha3S-xyl-Rha3S
*Formosa agariphila* KMM 3901	PL28	DP2,4	-
*Nonlabens ulvanivorans* NLR48	PL28	DP2,4	∆-Rha3S
*Nonlabens ulvanivorans* NLR42	PL28	DP2,4	∆UA-R3S-Xyl-R3S
*Formosa agariphila* KMM 3901	PL37	DP2,4	∆-Rha3S, ∆-Rha3S-Xyl-Rha
*Formosa agariphila* KMM 3901	PL40	-	-

**Table 2 molecules-27-03420-t002:** Samples used in this study and the list of isolates.

Samples (Sample Code)	Depth	Locater	Sample Date	Isolates
Ulva (uL)	0 m	34°44′32″ N, 136°31′48″ E	May 2018	ul20-2
Surface seawater (M)	0 m	34°44′32″ N, 136°31′48″ E	May 2019	MN2, MN3, MN-1, MN-2, 10N, 20G
Sediment MK6 point (D)	20 m	34°41′11″ N, 137°10′08″ E	May 2019	DN-1, DN-3, DG-3
Bottom seawater MK 1 point (**1**)	20 m	34°44′30″ N, 137°03′57″ E	May 2019	1G-1, 1G-1-2
Bottom seawater MK6 point (**6**)	20 m	34°41′11″ N, 137°10′08″ E	May 2019	6G-1

## Data Availability

The whole genome sequence has been deposited in the DDBJ/EMBL/GenBank under the accession number AP024154. Raw data have been deposited as DRX339462 and DRX339463. The BioProject number is PRJDB10725. The BioSample number is SAMD00254950 (*Echinicola* sp. 20G). The 16S rRNA gene listed in this study has been deposited as LC706867 to LC706879.

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
