# Peer review of "Isolation, Diversity and Characterization of Ulvan-Degrading Bacteria Isolated from Marine Environments"

_molecules, 2022, doi:10.3390/molecules27113420_

Round 1

Reviewer 1 Report

Manuscript deals on the molecular classification of ulvan degrading bacteria as well genomic analysis of some genes involved on.

Manuscript needs major and minor comments before to be accepted

Major

  1. Sequences of the glucosidases and 16S genes must be deposited in the Gene Bank and referred in this manuscript

Minor

  1. Do not use the term “strains, they are isolates please correct in the entire manuscript
  2. Please use italics for the genus names in the entire manuscript.
  3.  More details please see the attached manuscript

Reviewer 2 Report

The authors of the present manuscript isolated several ulvan-degrading bacteria from marine sediments and seawater. For some of the isolates ulvan-degradation was described first in this study. The ulvan degradation behavior for three of the isolates was studied in a little more detail. For one of the isolates (Echinicola), genome sequencing was carried out. However, the authors missed out to publish the genome publicly. In general, it is highly interesting to study ulvan degradation, especially with the aim to study the enzymes involved in this process. The research presented in this paper seems to be sound, but is lacking some more in depth studies. Furthermore, there is one misconclusion from the results, that needs to be addressed. Please see my following comments for more details.

Major comments:

p.5, 2.8: please deposit the genome and the raw reads in a publicly accessable repository (eg. NCBI) and add the accession number.

p.6, Fig.1, legend: The caption for Fig.1 B is misformulated. Better: "The appearance of a white color indicated presence of ulvan". Ulvan degrading activity cannot be indicated on such a test plate without bacteria. Instead, ulvan-degrading acitvity will be indicated by formation of a clear zone around a colony that is able to degrade ulvan (see fig. 3).

p.7, l. 211-213: I disagree. Fig. 4C shows an increase of absorbance. Please add a background measurement with any enzymes added. Only then you'll be able to judge, if the increase seen in 4C is really an activity or not.
Same for Fig. 5. 5A actually could be showing a very low increase.

p.10, l. 276-277: I disagree with the statement that the Echinicola enzyme is intracellular. It can also be periplasmic or cell surface associated bound to a cellulosome-like structure. It could even be bound lightly to the cell surface, since the cell pelllet wasn't washed before sonication according to the methods section. To further clarify this question, I suggest that the authors predict any possible N-terminal singal peptides for the enyzmes using SignalP to figure out if the enzyme likely is secreted or not. In parallel, the authors should look for potential C-terminal secretion signals as found in other algal polysaccharide-degrading enzymes (eg. https://www.frontiersin.org/articles/10.3389/fpls.2022.823668/full). 

General: Please link the observed activity in Echinicola to the genome by for example RNA-Seq. or qPCR. It would be even better to perform genome sequencing for all isolated strains and link expression of the discovered ulvan-related activities to the different genomes.

Minor comments:

p. 1, l. 12-14: I assume the authors mean that these genera were "isolated for the first time from Ulva sp." Otherwise the sentence is misleading, since these bacteria from these genera are in culture for a long time.

p.1, l. 14-15: Why is it important to produce an intracellular ulvan lyase? Ulvan is a polymer and is probably degraded outside of the cell to smaller pieces first, which are then transported across the membrane.

p.3+4, 2.1+2.2: please add GPS coordinates for the sampling sites.
p.4, 2.2: Listing the samples in a table would be better. Right now it is not well readable. Please also add some more info about the water mass sampled: salinity, temperature, oxygen, pH. What was the water depth the bottom mud sample was taken from? From which sediment depth was the bottom mud sample taken?

p.4, 2.4: please add more details here: time+temperature of incubation with CPC

p.4, 2.5: please add concentrations of primers, polymerase, buffer and dNTPs used for the PCR.

p4+5, 2.4+2.6: these sections seem to be redundant and can be combined

p.5, l.154: what was the cut-off and the membrane material of the used Amicon filter?

p.5, l.156: please add more details on the ultrasonic treatment. Specifically, add the settings of the instrument (eg. power, cycle).

p.5, l. 160-162: please add the temperature used for incubation in the plate reader.

p.5, 3.1: Title for this section is misleading. Anyway, This section shouldnt be part of the main text. I would suggest to transfer it into the supplement.

p.6+7, 3.2+fig.2: please break down the proteobacteria into the classes (Alpha-, Beta-, etc.). Proteobacteria is too broad.

p.7, l. 206: Figure 3 shows only a qualitative assays for ulvan degradation. Therefore, I suggest not use the word measured here.

p.7, l. 210-211: FIgure 4 shows just a reaction time course, not reaction kinetics.

p.8, fig.3: please add the strain names as in fig. 2 to the caption.

p.8+9, fig4+5: please combine these figures into one plot for each isolate and use different symbols for extracellular and intracellular. This increases readability of the paper.

p.9, 3.4: please add GH family numbers for all CAZymes in the clusters. Then the reader can relate much easier to these enzymes. If these clusters contain all enzymes necessary for the complete degradation of ulvan, I'd suggest to call them "polysaccharide utilization loci (PUL)" instead of just cluster. This term is generally well-known and recognized by the CAZyme community.

p.9+10, fig. 6+7: please label also all white and grey arrows with their annotations.

p. 10, discussion: It would be helpful for the dicussion, if the authors add a figure showing the degradation scheme of ulvan similar to ref. 4 and indicate in this scheme, which of the required genes have been identified in Echinicola. Currently, it is not clear whether a complete ulvan degradation pathway was found in the studied Echinicola genome or not.

p.10, l. 260-262: Why? Was there ulvan in the seawater? 
p.10, l. 262-264: Please mention GFC filter pore size again.

Round 2

Reviewer 1 Report

Manuscript can be accepted for publication